# Opportunities for Tailored Support to Implement Smoke-Free Homes: A Qualitative Study among Lower Socioeconomic Status Parents

**DOI:** 10.3390/ijerph17010222

**Published:** 2019-12-27

**Authors:** Els C. van Wijk, Regina I. Overberg, Anton E. Kunst, Janneke Harting

**Affiliations:** 1Amsterdam UMC, Department of Public Health, Amsterdam Public Health Research Institute, University of Amsterdam, Meibergdreef 9, 1105 AZ Amsterdam, The Netherlands; e.c.vanwijk@amsterdamumc.nl (E.C.v.W.); a.e.kunst@amsterdamumc.nl (A.E.K.); 2Department of Public Health, Public Health Service Kennemerland, Zijlweg 200, 2015 CK Haarlem, The Netherlands; ROverberg@ggdkennemerland.nl

**Keywords:** secondhand smoke, smoke-free home, parents, social class, qualitative research

## Abstract

With the aim of preventing children from being exposed to secondhand smoke, we examined to which level lower socio-economic status (SES) households had implemented home smoking rules and the factors that hampered parents in their process of change toward a complete smoke-free home (SFH). We conducted a qualitative study including semi-structured in-depth interviews with 14 parents of young children living in a lower SES neighborhood of a provincial town in the Netherlands. Interview transcripts were subjected to a qualitative content analysis. Three distinct levels of SFH implementation emerged: complete SFH, flexible SFH, and partial SFH. Differences between parents at these three levels essentially concerned: (1) the role of child-related moral considerations in their motivation for an SFH; (2) whether they felt they had the agency to set and enforce home smoking rules; (3) the difficulties they experienced in changing their smoking habit from smoking indoors to smoking outdoors. Parents also had different opinions about the role their children could play in facilitating the parental process of change. We conclude that the current level of SFH implementation may serve as a starting point for developing tailored interventions. Such interventions should probably address other factors than the commonly used awareness–knowledge–commitment approach.

## 1. Introduction

Worldwide, 40% of children are exposed to secondhand smoke [1]. The prevalence is even higher in Europe, where it varies between 51% and 61% [1]. Children’s exposure to secondhand smoke can have serious health consequences, such as sudden infant death syndrome, lower respiratory tract infections, asthma, and middle-ear disease [1,2,3]. As a consequence, 28% of the total child mortality can be attributed to exposure to secondhand smoke [1]. Hence, preventing children from being exposed to secondhand smoke could considerably improve their health [4]. Child exposure to secondhand smoke is most likely to happen at home [1,3], especially in lower socioeconomic status (lower socio-economic status (SES)) households in which one or both of the parents smoke [3,5,6].

A promising preventive action in this respect is the—voluntary—implementation of smoke-free homes (SFHs). Having an SFH implies that both family members and visitors are prohibited from smoking anywhere inside the home [4]. SFHs have been found to reduce the exposure of children to secondhand smoke, as well as the exposure of adult non-smokers, smoking rates among adults, and perhaps even smoking rates among youths [7]. Although the self-reported proportion of SFHs appears to be growing in recent years [8,9], the reported increase is likely to be an overestimation, as parents, especially those who smoke, are likely to both underestimate and underreport the actual exposure of their children to secondhand smoke [10,11,12]. Meanwhile, SFHs are still less prevalent in both lower-SES households and households that include at least one adult smoker [3,8].

A potential solution is to use interventions to support parents in the implementation of an SFH. Many of the current SFH interventions are school-based and offer information, education and/or a smoke-free promise for families to sign [13,14,15,16,17,18,19,20,21,22]. In some other interventions, smoke-free home advisors provide individual coaching or community-wide SFH promotion [18,19,20,21,23,24]. Few interventions also give parents information about smoke-related biomarkers in their children [23,24]. Although some of these interventions aim to strengthen the parents’ self-efficacy and skills to set and enforce home smoking rules [20,23], most aim to enhance the children’s abilities to strengthen the parents’ awareness, knowledge, and commitment concerning the risks and prevention of secondhand smoke [13,14,15,16,17]. A recent review concluded that only a minority of these interventions effectively reduced children’s exposure to secondhand smoke [4]. And although some interventions that targeted lower-SES households showed promising results, they tended to be less successful in smoking households [13,14,20,25]. For example, one study found an overall increase in SFHs from 35% to 68%, whereas in households with at least one smoker this proportion only increased from 41% to 48% [13]. Another study found similar—albeit not statistically significant—differences in success rates between non-smoking households and households with at least one adult smoker [21]. However, irrespective of the household’s smoking status, the features that make SFH interventions effective in facilitating parents to prevent their children from exposure to secondhand smoke are currently insufficiently known [4].

Examining the parental process of change that leads to an SFH could provide useful information [26]. A recent review summarized the previously identified facilitators and barriers to the implementation of SFHs [27]. These include: (1) knowledge, awareness, and risk perception (e.g., not knowing the risks of secondhand smoke); (2) agency and personal skills or attributes (e.g., lack of negotiation skills); (3) community and personal moral norms (e.g., feelings of guilt); (4) social influences (e.g., support from household members); (5) perceived benefits, preferences, and priorities (e.g., seeing smoking as stress management); (6) addiction and habit (e.g., unable to change indoor smoking habits); (7) practicalities (e.g., bad weather). However, these earlier studies have typically not focused on the relations between these factors and the process of change [26]. Nonetheless, evaluation studies of interventions to promote an SFH do suggest that implementing home smoking rules is a step-wise process [13,19,20,21]. That is, in agreement with implementation theories [28], households seem to progress from no home smoking ban, via a partial ban, to a full home smoking ban [19,21]. During this process of change, it is likely that parents have to alter their behavior step by step, meaning that a subsequent level of SFH implementation requires selected strategies targeting the specific factors that hamper change [29,30]. This would imply that the level to which households have implemented an SFH could serve as a starting point for the development of tailored interventions [31], which address a targeted selection of factors from the multitude of barriers previously identified [27].

In sum, especially in lower SES households, the proportion of SFHs is low [3,5,6], while the success of SFH interventions remains limited [13,14,20,25]. Hence, it is particularly important to prevent children in lower SES households from exposure to secondhand smoke [1,12]. Therefore, the aim of our qualitative study was to examine the possibilities for tailoring SFH promoting interventions to the level to which home smoking rules have or have not been implemented by lower-SES households. We aimed to answer the following research questions: (1) To what level did parents implement home smoking rules? (2) Which factors influenced the process of setting and enforcing home smoking rules? (3) How were these factors related to the level of implementation of an SFH?

## 2. Materials and Methods

### 2.1. Design

Between May and July 2017, we performed a qualitative study in which we held semi-structured in-depth interviews with smoking and non-smoking parents living in a lower-SES neighborhood of a provincial town in the Netherlands. In 2016, this neighborhood scored 9 on an SES indicator scale (including levels of education, income, and occupation) ranging from 1 (high SES) to 10 (low SES).

### 2.2. Sample and Recruitment

The inclusion criteria for parents were: being 18 years or older, having at least one child aged 0–14 years, being part of a household in which at least one adult smokes or has smoked in the past, or with regular visits by smokers, being able to speak and understand Dutch, and having no crisis situation in the family, such as being mixed-up in a divorce, as to avoid family tensions arising or intensifying. Parents were recruited within their own neighborhood, at schools, community centers, and local events (e.g., summer festival). The main recruiters were the second author [R.I.O.] and other officials from the Regional Public Health Service. They were supported by local health care providers and social workers. The best recruitment strategy appeared to be person-to-person talks in which local residents were invited to participate in a study about smoking in the home environment without raising the subject of SFHs. These talks were supported by flyers with information about the study. About 100 local residents had to be approached to reach a sufficient sample size.

### 2.3. Instruments and Procedure

Interviews were conducted by the second author [R.I.O.] at parents’ homes or local community centers. Parents signed an informed consent form before the start of the interview. Interviews took approximately one hour, and addressed topics such as family composition, living condition and smoking behavior, current home smoking rules, and the motivation for, intention to, and process of implementing an SFH (see Table 1). The interview guide was based on empirical knowledge, such as the previously identified facilitator and barriers [27], and theoretical insights, like from behavior change and implementation theories [30,31]. At the end of the interview, each respondent completed a short questionnaire assessing socio-demographic characteristics. After the interview, respondents received a gift voucher of 15 Euros. The interviews were audio-recorded and transcribed verbatim.

### 2.4. Analysis

Transcripts were subjected to a qualitative content analysis [32] in MaxQDA 12 [33]. We developed a thematic framework using deductive and inductive coding [32]. Deductive codes were based on the interview guide and inductive codes were derived from insights obtained during the analysis.

The first three transcripts were coded independently by two authors [EvW and R.I.O.]. They discussed their differences regarding the coding of these transcripts and memos about newly emerging themes (e.g., relating to the level of SFH implementation) until consensus was reached. For some themes (e.g., ‘bottlenecks’ and habit), new sub-codes were added to the thematic framework for further analysis. The remaining transcripts of the interviews were independently coded by one of the authors. They checked each other’s coded transcripts and memos and discussed disagreements until they reached consensus. The definitions of codes and sub-codes were continuously updated during this process.

In the subsequent cross-case analysis, the first and last authors [E.C.v.W. and J.H.] compared the levels to which respondents had implemented an SFH by reading through the full transcripts again while taking into account the memos and coding. Three different implementation levels emerged: complete SFH, flexible SFH, and partial SFH. For each of these three levels, the first author [E.C.v.W.] summarized the codes and sub-codes for home smoking rules, and the motivation for, intention to, and process of implementing an SFH in three separate matrices (Appendix A). The comparative analysis and matrices, providing the basis for the results section below, were checked and agreed with by the second and last authors [R.I.O. and J.H.].

### 2.5. Ethics

According to the Dutch Medical Research Involving Human Subjects Act, this study did not require approval by a medical research ethics committee in the Netherlands. However, the research proposal was reviewed by the independent funder of the study. In addition, the procedure to obtain informed consent underwent internal peer review. We followed the ethical principles for medical research involving human subjects, as laid down in the Declaration of Helsinki and adopted by the World Medical Association.

## 3. Results

### 3.1. Sample Characteristics

The study sample (*n* = 14) included five parents with a complete SFH, three with a flexible SFH, and six with a partial SFH. Most respondents were female (*n* = 13) and smoked (*n* = 8). The mean age of the respondents was 38.8 years (SD = 5.0), the mean age of their children 8.3 years. Ten respondents had a partner, most of whom smoked (*n* = 6), about half had a paid job (*n* = 8). The majority lived in a single-family home (*n* = 10), while the others inhabited an apartment (*n* = 4).

### 3.2. Complete SFH

Most respondents with a complete SFH were non-smokers (Appendix A), who typically had very clear home smoking rules. Smoking was not allowed indoors, only in the garden or at the back of the garden, on the balcony, in front of the front door or in the shed.


*“I really told them, if there’s going to be any smoking, then it’s outside. And I also do the same with birthdays and things like that, people are allowed to smoke outside, or in the shed if the weather is really bad, but I tell them smoking is no longer allowed indoors.” *
(I10; smoker; single parent)

The start of an SFH had always been part of a life transition, due to, for instance, pregnancy, childbirth, moving house or renovations, while the parents’ primary motivation for an SFH was the health and well-being of their children.


*“The fact that my children would inhale smoke, that will bother them. I mean, as a mother, I don’t want that for my children, I just want them [...] to live in a healthy home.”*
(I14; non-smoker; smoking partner)

In addition, smoke was perceived as ‘filthy’.

Parents with an SFH tended to strictly enforce the smoking rules, in which they succeeded by high levels of assertiveness and willpower, despite initial resistance among smokers and the risk of losing touch with friends or family.


*“I used to have some friends when I was still living in that flat, and I actually lost them because they found it [the implementation of an SFH] ridiculous. I said: ‘Why do you think it’s ridiculous? It’s my life. After you’ve left I’m sitting there in the stench’ [...] But I do think that when you have children, you have to take that into account. You want your children to grow up healthy.” *
(I05; non-smoker; smoking partner)

Most smoking family members and friends accepted the rules after a while, and now they all automatically went outside to smoke. Barriers to the implementation of an SFH were hardly reported. Facilitators were related both to the built environment (e.g., a home without proper ventilation) and the social environment (e.g., no other smokers living in the household; smoking visitors who as a rule ask where they are allowed to smoke).


*“Nowadays, everyone just asks if they may smoke inside, and then it’s also very easy to say: ‘no, there’s no more smoking indoors’.” *
(I10; smoker; single parent)

In households that included at least one smoker, respondents experienced some additional advantages of an SFH, such as a healthier living environment and less smoking by the smoker.


*“When you smoke indoors it’s very easy to light a cigarette. Now you have to make the effort because you have to go all the way outside, so then you’re going to smoke less, and that was also a reason for me, I thought like, you know, that I would smoke less.” *
(I10; smoker; single parent)

Few respondents identified a role for their children in the implementation of an SFH, such as putting pressure on parents, reminding them of the rules, or setting home smoking rules together.


*“I think [as a parent] you’re more confronted with the arrangements you make with your children.” *
(I05; non-smoker; smoking partner)

### 3.3. Flexible SFH

Although all respondents with a flexible SFH smoked (Appendix A), they too did not allow smoking indoors, only outdoors, in the garden, but most of them applied these rules in a more flexible way. They were more inclined to allow exceptions (e.g., smoking indoors in case of late visitors or during parties if the weather is bad) and to violate the rules (e.g., walking into the house with a cigarette to get something).


*“We used to smoke inside the home before we got children. In principle, we still don’t smoke inside the home. It happens sometimes, for instance when we have visitors or have a party or things like that.” *
(I08; smoker; smoking partner) 

For these parents too, the start of an SFH had been part of a life transition, but their primary motivation originated from actual or anticipated feelings of guilt. Children do not choose to be exposed to smoke, and parents were afraid to be blamed for it if their children became ill in the future.


*“But at any rate I don’t want her to say, later, yes my mother used to smoke, and now I have this. That it’s my fault. That I feel that it would be my fault if she gets ill.” *
(I12; smoker; smoking partner) 

Additional motives for implementing an SFH were health or wellbeing in general and a clean and fresh-smelling house.


*“It’s also for the children, but also because I just find it’s filthy inside. Everything stinks and turns yellow, I think it’s filthy. It means I don’t have a fresh-smelling house.” *
(I11; smoker; smoking partner) 

Parents with a flexible SFH were less passionate and assertive, and thus more relaxed in their discussion and enforcement of the home smoking rules.


*“What has occasionally happened, is that when we’re sleeping, the children and I, and my friend is still awake, and he gets a visitor, then they sit here at the table to talk, and then they sometimes light a cigarette, but it’s not a rule or a habit. […] I don’t have so many problems with it, although I still think it’s a pity that it happens. […] It’s not such a problem.” *
(I08; smoker; smoking partner)

Although the rules were accepted, the smoking parents had found it quite hard to break their own strong habit of smoking indoors, for instance on the couch while watching TV. Although it took some time, they eventually succeeded in changing this habit to mostly smoking outdoors.


*“We didn’t quit smoking indoors from one day to the next [...] In the beginning, we also occasionally lit a cigarette when we were watching TV in the evening and the children were already sleeping upstairs. But at a certain point, that was no longer necessary, because you grow accustomed to it [only smoking outdoors]. You know that you’re breaking the rule when you smoke inside. You gradually phase it out, and now I can say for myself when I watch TV: ‘I don’t even want to smoke’.” *
(I08; smoker; smoking partner) 

A typical facilitator in the built environment was having a comfortable outdoor place to smoke. Other facilitators were having the necessary willpower, removing ashtrays from the home, and planning a farewell ritual (e.g., including celebration and cleaning activities).


*“It was, like, this is the last night, now smoking is still allowed in the house, and after today it’s finished. Tomorrow I’m going to wash the curtains, wash the pillows. And that’s how it went.” *
(I12; smoker; smoking partner)

A common barrier to smoking outdoors was bad weather. However, in combination with the home smoking rules, bad weather conditions could also serve as an encouragement not to smoke—at least temporarily.


*“Then I just don’t smoke [when it is raining], unless that monster [smoking addiction] should be so bad and it’s dark in the evening, then I would sometimes secretly smoke at the front door, but I think that’s no longer for the fun of it, so you better just suppress that monster and just go to bed.”*
 (I12; smoker; smoking partner)

Regarding the advantages experienced, for parents with a flexible SFH having a clean house seemed to be more important than a healthy home environment.


*“First smoke outside, and then decide to quit. But if you smoke outside, you can already start cleaning your house, and you will notice how fresh it smells.” *
(I12; smoker; smoking partner)

Parents with a flexible SFH also suggested a greater variety of roles for their *children*, such as supporting their parents in implementing home smoking rules.


*“But they can involve their children [in programs to implement SFHs]. Give them the task to support mummy and daddy and to help their parents to remember or... Sure they’ll like that.” *
(I11; smoker; smoking partner)

Nevertheless, implementing an SFH was seen as basically the responsibility of the parents.

### 3.4. Partial SFH

Most respondents with a partial SFH (Appendix A) were smokers who had set “kitchen rules”, meaning that smoking indoors was only allowed in the kitchen.


*“There’s simply no smoking in the house, except for the kitchen, but nowhere else, not in the living room.” *
(I07; non-smoker; smoking partner)

Although their primary motivation for these rules was also related to children, pregnancy, and childbirth had been a far less important stimulus. Also, the child-related motivation was often much less specific than the other motives, including the dirtiness of smoking, a new house, having puppies, being a non-smoker, or having a non-smoking partner.


*“Because my partner doesn’t smoke, and he says: ’I am in your smoke, and it stinks, and everything here smells, the living room, my clothes’. I’m a smoker, so I do not smell it that much, but for someone who doesn’t smoke, they notice it. And you also notice it in the morning that your living room is just not smelling very fresh when you’ve smoked.” *
(I06; smoker; non-smoking partner)

They also had less pronounced motives to implement an SFH. These motives included their own health and that of others, as well as the smell of smoke and a lower frequency of smoking.


*“Yes, actually I want to [have an SFH]. […] Because it’s just better and healthier for my environment. […] Maybe I’ll be smoking less, that would be nice too.” *
(I06; smoker; non-smoking partner)

Parents with a partial SFH were much more likely to violate their home smoking rules. This mostly happened when the children were away or asleep.


*“Especially when my son goes to his father every other weekend, and then he’s not here and then I tend […] to open all windows and doors and then I can smoke in the living room. That’s of course not what was intended. But that happens sometimes. Then I think, well, it’s easy, he’s not there, as if suddenly I’m doing it [only smoking in kitchen] just for him.” *
(I06; smoker; non-smoking partner) 

The many exceptions and violations were partly the result of having unclear rules. This lack of clarity made it difficult to enforce the rules. In addition, both smoking and non-smoking parents felt that they were not in a position to initiate, communicate, reinforce, or strengthen home smoking rules.


*“Obviously, having a smoke-free house would be nice. But I’m not the one who smokes. So then it’s difficult to set the rules. I can’t decide for him [partner who smokes] and say to him: ‘From now on you go outside or you quit smoking’. You just can’t say that.” *
(I07; non-smoker; smoking partner)


*“It’s a bit of a concession [having a partial SFH]. […] For me as an occasional smoker […] I find it a little bit hypocritical to make the rules even stricter.” *
(I13; smoker; smoking partner)

Characteristic was the parents’ struggle to change their habit of smoking anywhere in the home to a habit of smoking only in the kitchen. This habit change was perceived as a big step, since compared to the living room, the kitchen was seen as a much less comfortable place to smoke. Still, most parents viewed an SFH as the next step, albeit a difficult one.


*“I’m not in my kitchen for nothing, otherwise I would have sat on the couch. You know it’s not good. So why not take that step further? […] I also find that very difficult.” *
(I02; smoker; single partner)

Perceived barriers to realizing a complete SFH were especially related to the built environment: having no outdoor space (e.g., no garden or balcony), having no comfortable or safe outdoor place to smoke, and bad weather.


*“And then in the evening, [...] it also feels a bit scary to smoke in the garden as a single parent. […] I think it’s just uncomfortable to smoke outside in the evening.” *
(I02; smoker; single parent)

Perceived facilitators related to social support (e.g., support from an acquaintance, conversations with the partner or reminders by the children), skills and self-efficacy (e.g., increased assertiveness to set and enforce rules), and the built environment (e.g., shelter, garden lighting, and a bench for outside).


*“If I had such a shelter, so you’re dry outside, that would of course help a lot [to implement an SFH]. I’m indeed thinking about the evening hours.” *
(I02; smoker; single parent)

At the same time, implementing an SFH was perceived as unrealistic, because changing the habit of smoking indoors, e.g., to relax or socialize, to smoking outdoors would be even more uncomfortable than smoking only in the kitchen. Therefore, most parents thought they could just as well quit smoking.


*“Because then it [implementing an SFH] would be the same as if he would quit smoking. We live in an upstairs apartment. […] We don’t have a balcony or a garden, so we really have to go downstairs and then smoke in front of the front door. That also looks a bit weird. But no, that’s just not convenient, so we never actually talked about it.” *
(I07; non-smoker; smoking partner)

Most parents with a partial SFH saw a substantial role for their children in the implementation of an SFH, such as confronting parents with their smoking behavior, reminding parents of rules, determining a reward for implementing an SFH, and supporting parents in smoking cessation. This is what some children actually already do.


*“I’ve been told by my son that I must smoke outside. […] And then I will consider it. […] He says: ‘You’re going to die of smoking and if you want to smoke that’s your own concern. But I don’t want it in my presence’, so could I just go outside.” *
(I06; smoker; non-smoking partner)

However, not every parent agreed with giving children such a role, as they thought it was the smokers themselves who were responsible.

## 4. Discussion

### 4.1. Summary of Findings

In our qualitative study among lower SES parents, we noticed three levels of SFH implementation: complete SFH, flexible SFH, and partial SFH. In addition, we found that the factors influencing the process of creating an SFH were likely to be related to the level to which the household had currently implemented home smoking rules. Between these three implementation levels, parents essentially differed regarding: (1) the role of child-related moral considerations in their motivation for an SFH; (2) whether they felt they were in the position to set and enforce home smoking rules; (3) the difficulties they experienced in changing their habit from smoking indoors to smoking outdoors. Finally, we saw differences in the parents’ opinions about the role children could play in facilitating the process of change leading to an SFH.

### 4.2. Interpretation

The factors we identified largely correspond to those reported in a recent review study [27]. However, they substantially differ from those addressed by the awareness–knowledge–commitment approach that was commonly used by previous SFH interventions [13,14,15,16,17]. This may mean that future interventions should target different factors in order to support lower SES parents in adopting and carrying our their own actions and strategies leading to an SFH, such as cleaning the house, establishing, negotiating, and reinforcing clear home smoking rules, and creating a comfortable enough outdoor smoking place. Below we discuss how our findings may be used to develop tailored SFH interventions to support lower SES parents in their efforts to prevent their children from being exposed to secondhand smoke.

#### 4.2.1. Strengthening Parental Motivation

In our study, parents who differed in the level to which they had implemented home smoking rules also differed in their moral considerations regarding an SFH. For parents who had realized a complete SFH, the primary motivation had been their parental duty to protect the child’s health and wellbeing. In line with the self-determination theory (SDT) [34,35], this motivation reflects a person’s core values [35], such as the intrinsic personal norm of being a good parent [36]. Parents with a flexible SFH tended to primarily act out of feelings of guilt toward their children, which SDT would regard it as an expression of the extrinsic subjective norm [35], i.e., as a moral obligation based on external pressure [36]. Parents with a partial SFH did, in terms of SDT, not act out of moral considerations but from attitudinal considerations [35]. That is, their home smoking rules were not based on their children’s health and wellbeing, but on instrumental and affective consequences [36], like the dirt smoke causes indoors and the inconvenience of smoking outdoors.

Notably, most SFH interventions we know of have tended to focus on the parent’s attitudinal motivation in terms of increased awareness, knowledge, and commitment regarding the risks and prevention of child exposure to secondhand smoke [13,14,15,16,17]. Our results indicate that progressing through the stage-wise process of implementing an SFH may additionally require normative-oriented intervention approaches. Such approaches may, for instance, appeal to feelings of guilt, e.g., by increasing risk perception [37,38], improving perceptions of child exposure [10,11,12], and eliciting anticipated regret [39]. Such interventions may alternatively aim to strengthen the parents’ moral norms, e.g., by appealing to core values relating to good parenthood [36]. In doing so, one should realize that direct appeals may elicit reactance [39], and that indirect appeals, making use of parents as role models [20,21], may be more appropriate [39], also in order to avoid unwelcome stigmatization [27]. In this respect, normative-oriented interventions should also make sure that they additionally provide parents with the agency needed to deal with the many challenges they are facing [27].

#### 4.2.2. Strengthening Personal Agency

In our study, parents without a complete SFH often felt they were not in a position to set and enforce home smoking rules. This may relate to the concept of personal agency, meaning that “people bring their influence to bear on their own functioning and on environmental events” [40]. The parents’ reports in our study illustrate how personal agency includes both perceived self-efficacy and skills [27,41]. For example, parents reported that they found it hard to stick to their own home smoking rules or to discuss these roles with other, mostly smoking, household members. This finding may indicate the presumed importance of “family talk” in interventions [26].

So far, most interventions to support parents in the implementation of an SFH have aimed at strengthening the children’s, rather than the parents’, self-efficacy and skills, although we are aware of some exceptions [20,21,22,23]. In order to improve the parents’ self-efficacy and skills, interventions for parents with a partial SFH could, for instance, make use of enactive learning [31]. This could involve practicing negotiation skills to discuss an SFH with household members in a training situation, such as role-play [26]. Interventions tailored to parents with a flexible SFH would rather have to strengthen assertiveness and self-discipline, for example through a training program [42,43], but these factors remain hard to change.

Whether children should indeed have a role in strengthening parental agency, as was also suggested by parents in our study, seems questionable. Although previous studies illustrated the feasibility of such a role [13,14,15], most parents in our study also questioned the children’s responsibility in this respect. As personal agency generally seems to be beyond the sphere of influence of children, their role in this respect deserves careful consideration.

#### 4.2.3. Transforming Parental Smoking Habits

Our results confirm the earlier finding that the habit of smoking inside the home is hard to break [26,27,44]. The smoking parents in our study had struggled, or were still struggling, with changing their habit of smoking indoors to smoking outdoors. For parents with a flexible SFH, who already frequently smoke outside the house, this initial habit change could benefit from relapse prevention strategies, such as problem-solving to overcome remaining barriers [26], and positive reinforcement strategies, such as creating public commitment by signing a contract [13,20].

However, just as in other studies [26,27,44], weather conditions, along with the inconvenience of going outside, remained a major barrier for parents with a partial SFH. This indicates that this specific change in smoking habit may require environmental restructuring [31], not just inside the home [24], but especially outside. That is, households that have an outdoor space available could change it in such a way that it facilitates smoking outdoors, e.g., by creating a comfortable and safe outdoor smoking place. We also found that going outdoors may mean that smoking loses its value in terms of relaxation and socialization, so these aspects may also deserve attention.

If an acceptable outdoor space is lacking, our findings indicate that parents may feel they could just as well quit rather than set more restrictive home smoking rules. Although quitting is very difficult for lower SES smokers [45], our results indicate that SFH interventions should always be accompanied by an offer to support a quit attempt.

### 4.3. Limitations

Our study was subject to some limitations. First, it was a cross-sectional qualitative study [32]. Our results indicate that parents encounter different influential factors in different stages of the process of implementing an SFH. However, our design did not allow us to follow the parents’ actual progress through the stages. Therefore, a longitudinal study would be needed [32].

Second, due to our small sample size, we may not have reached data saturation for all three levels of SFH implementation [32]. Also, the sample size did not allow for cross-case analyses on factors other than the level of SFH implementation. Therefore, other factors than those reported here may be at play in the implementation process. As including lower SES parents in a study usually takes a lot of effort and time, including a larger sample would have exceeded the study’s resources.

Third, our sample is presumably a selected group, that is not representative in every respect. For instance, most respondents were women. Also, the sample did not include parents without home smoking rules [46], nor did our parents talk about difficult social and economic circumstances [27]. Although we were able to include parents living in a lower SES neighborhood, we did not collect data about income, as to avoid resistance and drop-out from the study. Also, our respondents may have already been interested in the topic and also relatively well-off. Follow-up studies could benefit from recruitment by community members or community leaders [47].

## 5. Conclusions

If we want to prevent children from being exposed to secondhand smoke, interventions to support parents in creating a smoke-free home (SFH) could start from the current level to which households have already implemented home smoking rules. Factors influencing the process of parental change toward a complete SFH differed between implementation levels, which provides opportunities for tailored intervention development. Such tailored interventions should most probably take a different approach than the awareness–knowledge–commitment SFH interventions commonly used so far.

## Figures and Tables

**Table 1 ijerph-17-00222-t001:** Interview guide.

Interview Topics Main codes	Examples of Interview Questions
Family composition, living condition and smoking behavior Household situation Type of house Smoking status	Can you tell me who you are? With whom are you living in your home? Could you please describe the house you are living in? Does any of the persons you live with smoke? If yes: Who smokes? Do your friends, acquaintances and relatives smoke? Does anyone smoke inside your home? If yes: Where in the home? Who? When?
Home smoking rules (HSRs) Introduction of HSRs Initiator of HSRs Content of HSRs Acceptation of HSRs Enforcement of HSRs	What are the smoking rules in the home? Who initiated the rules? When were the rules initiated? Are there exceptions to the rules? If yes: In what situation? To what extent are your partner and/or children satisfied with the rules? Are the rules sometimes violated? If yes: By whom? How often does that happen? In what situation? How easy or difficult is it to enforce the rules? Would you like to receive support to enforce the rules? If yes: By whom and how?
Motivation, intention and implementation process SFH Intention to have SFH Motivation for HSRs Smoking habits and HSRs Barriers to SFH Facilitators of SFH Extra advantages of SFH Role of children in SFH	[*Questions may depend on the level to which an SFH is implemented*] Why did you implement a smoke-free home? How did you do this? What enabled you to do so at that time? What made it difficult? Why did you implement a few smoking rules in the home? How did you do this? What enabled you to do so at that time? What made it difficult? Have you ever considered not smoking anywhere in the house anymore? If yes: Why did you consider this? Why have you not yet implemented a smoke-free home? Are there things that make it difficult? If no: Why not? What could be reasons to consider a smoke-free home? Would you like to have a smoke-free home right now? If yes: Why do you want that? What could help you to implement a smoke-free home? How do you perceive your own role and the role of your partner and/or children in implementing a smoke-free home? Would you like to receive support to implement a smoke-free home? By whom and how? If no: Why not?

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
