# Peer review of "Opportunities for Tailored Support to Implement Smoke-Free Homes: A Qualitative Study among Lower Socioeconomic Status Parents"

_ijerph, 2019, doi:10.3390/ijerph17010222_

Round 1

Reviewer 1 Report

This study explores the implementation level of the home smoking roles and the possible challenges that hinder the setting and implementation of the SFH, and in the end leads to the suggestion on improving the SFH.

The study design is fine and the manuscript is well structured, and the study is quite important for the design and implementation of smoking free home. I have few concerns in general.

In introduction section, I would suggest the author include a description on the current SFH interventions, such as what types. What does the author mean lower SES in this context? The sample size is quite small as the limitation indicated, are these samples representative? The author needs to clarify. In results and discussion section, I did not see much description related to the characteristics of the samples in Table 2. For example, if there are any differences between the smoking status/partner’s smoking status/ household situation etc. in implementing SFH? This could further help to facilitate tailored support for the implementation of SFH.

Reviewer 2 Report

In this qualitative study, 14 parents of a low SES status and with children under the age of 14 were interviewed regarding their smoking behaviors in their houses. Overall, parents indicated that they were motivated to have smoke-free homes to create a fresh and healthy environment for their children. Five parents were able to establish clear rules about smoking outside the house, three have designated areas in the house to smoke (e.g., the kitchen), and six, although having some rules for smoking only in designated spaces, they were not able to implement the rules in a consistent manner. The study addresses an important topic since second hand smoke is a big public health problem, especially when it affects children. Reducing smoke in the house would be highly beneficial for children but also to the parents which may reduce their smoking or even decide to quit smoking altogether. However, there are some limitations that reduce the quality of the results, such as a sample mainly composed by women, data saturation was not achieved, and everybody in the sample had home smoking rules. Below are also some concerns that could be addressed by the authors to strengthen the manuscript.

Introduction:

The study is focused on the population with low SES. In the introduction it is briefly mentioned that people with low SES are less likely to have a smoke-free home and that interventions that targeted lower SES households tended to be less successful in smoking households. However, since the study targeted only low SES individuals I think the introduction should emphasize more why focusing only on this segment of the population is important. At the end of the introduction it is indicated that “earlier studies have examined facilitators and barriers to the implementation of SFHs” (p. 2, line 64). Please specify which facilitators and barriers were already examined and what were their findings and conclusions.

Methods:

The sample was composed of smoking and non-smoking parents living in a lower-SES neighborhood. Do you have data about their annual household income? If not, this should be mention in the limitations. Were the parents selected based on the three levels of SFH? One of the inclusion criteria was “having no crisis situation in the family,” can you please explain or describe in more detail what does this mean? The places where participants were recruited are described but not how they were recruited. Please indicate the strategies used for recruiting the sample (e.g., flyers, announcements). In the instruments and procedures section it is indicated that one of the topics assessed in the interviews was “living situation.” What does living situation mean? The interview guide was based on theoretical and empirical insights. Which theoretical and empirical insights? The interviews were transcribed and then coded by two authors. What was the degree of agreement between coders? Can you please provide the Kappa index and overall reliability reached between coders? Authors indicated that three different implementation levels emerged from the interviews. However, it seems that this was not something that emerged from the data since the questions from the interview were already customized by those three categories (see table 1 of the manuscript).

Results:

Please describe the recruitment process: How many potential participants were screened? How many met inclusion criteria? How many agreed to participate in the study? The way the results are presented seems very lengthy and repetitive. I suggest organizing them by responding to the study aims rather than by the type of smoke-free home: to what level did parents in the sample implement home smoking rules; which factors influenced the process of implementation; and how were these factors related to the level of implementation. This will help the manuscript to flow better and to keep it consistency between aims and results. I think table 2 is not needed and the information could be summarized in a paragraph. Otherwise, I think adding percentages would help to better understand the table.

Discussion:

The discussion is also very lengthy and difficult to follow. The authors missed the opportunity to use the information gathered in the results to suggest possible strategies that could be used in a future intervention. For example, from the results it seems that life changes (e.g., pregnancy, childbirth) are important moments to start a smoke-free home. Common barriers are the lack of a space to smoke outside, bad weather, and smoking dependence. Strategies that seem to be working are establishing clear rules on where to smoke, communicating those rules to visitors in a clear manner, establishing a place to smoke (e.g., balcony, garden, street), cleaning the house and removing ashtrays and lighters, leaving cigarettes only in the smoking designated areas. These are all examples of strategies that were successful for those participants who were able to have a smoke-free home. If presented in an organized manner they could be useful for those who were not able to completely shift into a smoke-free environment or those who are thinking about it.

Round 2

Reviewer 2 Report

The authors were responsive with all my comments and suggestions and most of the suggested changes were included in the current version of the manuscript.